# Bioelectrical Impedance Analysis (BIA) for the Assessment of Body Composition in Oncology: A Scoping Review

**DOI:** 10.3390/nu15224792

**Published:** 2023-11-15

**Authors:** Mariana Garcia Branco, Carlota Mateus, Manuel Luís Capelas, Nuno Pimenta, Teresa Santos, Antti Mäkitie, Susana Ganhão-Arranhado, Carolina Trabulo, Paula Ravasco

**Affiliations:** 1Centre for Interdisciplinary Research in Health (CIIS), Universidade Católica Portuguesa, 1649-023 Lisbon, Portugal; cmmateus96@gmail.com (C.M.); luis.capelas@ucp.pt (M.L.C.); npimenta@esdrm.ipsantarem.pt (N.P.); sarranhado@uatlantica.pt (S.G.-A.); 2Nutrition and Dietetics, Hospital de Cascais Dr. José de Almeida, 2755-009 Alcabideche, Portugal; 3Faculty of Health Sciences and Nursing, Universidade Católica Portuguesa, 1649-023 Lisbon, Portugal; tsantos@ucp.pt (T.S.); carolinafptrabulo@gmail.com (C.T.); pravasco@ucp.pt (P.R.); 4Polytechnic Institute of Santarém, Sport Sciences School of Rio Maior, 2040-413 Rio Maior, Portugal; 5Interdisciplinary Centre for the Study of Human Performance, Faculdade de Motricidade Humana, Universidade de Lisboa, 1495-751 Lisbon, Portugal; 6Universidade Europeia, 1500-210 Lisbon, Portugal; 7Department of Otorhinolaryngology-Head and Neck Surgery, Helsinki University Hospital, University of Helsinki, 00014 Helsinki, Finland; antti.makitie@helsinki.fi; 8Research Program in Systems Oncology, Faculty of Medicine, University of Helsinki, 00014 Helsinki, Finland; 9Division of Ear, Nose and Throat Diseases, Department of Clinical Sciences, Intervention and Technology, Karolinska University Hospital, Karolinska Institute, 17176 Stockholm, Sweden; 10Atlântica, Instituto Universitário, Fábrica da Pólvora de Barcarena, 2730-036 Barcarena, Portugal; 11CINTESIS, Centre for Health Technology and Services Research, 4200-450 Porto, Portugal; 12Medical Oncology, Centro Hospitalar Barreiro-Montijo, 2830-003 Barreiro, Portugal; 13Católica Medical School, Universidade Católica Portuguesa, 2635-631 Rio de Mouro, Portugal; 14Centre for Interdisciplinary Research in Health Egas Moniz (CiiEM), 2829-511 Almada, Portugal

**Keywords:** cancer, body composition, bioelectrical impedance analysis (BIA), phase angle

## Abstract

Bioelectrical Impedance Analysis (BIA) is a reliable, non-invasive, objective, and cost-effective body composition assessment method, with high reproducibility. This scoping review aims to evaluate the current scientific and clinical evidence on BIA for body composition assessment in oncology patients, under active treatment. Literature search was conducted through MEDLINE, CINAHL, Scopus and Web of Science databases, following PRISMA-ScR Guidelines. Inclusion criteria comprised studies reporting the use of BIA for body composition evaluation in adults with cancer diagnosis. Studies including non-cancer pathology or only assessing nutritional status were excluded. This scoping review comprised a total of 36 studies: 25 were original studies including 18 prospective studies, six cross-sectional studies and one retrospective study and 11 were systematic reviews. Population size for the included original articles ranged from 18 to 1217 participants, comprising a total of 3015 patients with cancer with a mean baseline Body Mass Index (BMI) ranging from 20.3 to 30.0 kg/m^2^ and mean age ranging between 47 and 70 years. Review articles included a total of 273 studies, with a total of 78,350 participants. The current review considered studies reporting patients with head and neck cancer (HNC) (*n* = 8), breast cancer (BC) (*n* = 4), esophageal cancer (EC) (*n* = 2), liver cancer (*n* = 2), pancreatic cancer (PC) (*n* = 3), gastric cancer (GC) (*n* = 3), colorectal cancer (CRC) (*n* = 8), lung cancer (LC) (*n* = 1), skin cancer (SK) (*n* = 1) and multiple cancer types (*n* = 6). BIA is a suitable and valid method for the assessment of body composition in oncology. BIA-derived measures have shown good potential and relevant clinical value in preoperative risk evaluation, in the reduction of postoperative complications and hospital stay and as an important prognostic indicator in persons with cancer. Future research on the diagnostic value and clinical applications of BIA and BIA-derived phase angle (PhA) should be conducted in order to predict its impact on patient survival and other clinical outcomes.

## 1. Introduction

Nutritional deterioration and progressive unintentional weight loss are prevalent conditions in patients diagnosed with cancer, and are widely associated with poor outcomes and complications through the course of the disease and treatments [1,2].

Cancer leads to metabolic alterations that contribute to depletion in nutritional status resulting in changes in body composition, which in turn are negative predictors of therapy toxicity, clinical outcomes, quality of life and survival. Therefore, it is critical to timely identify and treat malnutrition in order to enhance clinical outcomes. Thus, body composition should be part of nutritional assessment in these patients. Nevertheless, it remains a challenge due to a variety of methods and tools to assess nutritional status and body composition in patients with cancer [1].

Methods for body composition assessment include anthropometry, bioelectrical impedance analysis (BIA), air displacement plethysmography (ADP), dual-energy X-ray absorptiometry (DXA), computed tomography (CT) and magnetic resonance imaging (MRI) [1,2]. While DXA has traditionally been regarded as the “gold standard” for assessing body composition, research has also emphasized the validity and reliability of body composition evaluation through CT and MRI [3,4]. Nonetheless, these methods are costly, may involve radiation (CT), and are not commonly performed, rendering them often inaccessible [5]. In contrast, Bioelectrical Impedance Analysis (BIA) emerges as a dependable, non-invasive, objective, and cost-effective alternative with high reproducibility and minimal training requirements [1]. Recent studies have even indicated potential advantages of utilizing BIA for body composition assessment in patients with cancer [6,7].

BIA estimates total body water (TBW) by measuring impedance (Z), which results from resistance (R) and reactance (Xc) components [1,8]. This relationship can be expressed using the equation Z^2^ = R^2^ + Xc^2^ [9]. R represents the body’s opposition to the flow of an alternating current, while reactance gauges the electrical charge stored in cell membranes. Impedance measurements can be obtained using single or multiple current frequencies [1,10]. BIA calculates an individual’s resistance to a weak electric current, enabling the estimation of a two-compartment model of body composition, which includes fat mass (FM) and fat-free mass (FFM), using empirically derived equations [5,11]. Lower BIA-derived FFM has been associated with malnutrition in hospitalized patients and linked to unfavorable clinical outcomes, such as prolonged hospital stays and increased 28-day mortality in intensive care settings [12,13,14].

In the context of medical oncology, BIA-derived body composition measures can be clinically relevant. Two crucial BIA-derived parameters are Bioelectrical Impedance Vector Analysis (BIVA) and Phase Angle (PhA), both of which provide insights into nutritional and hydration status [15]. BIVA assesses hydration and cell mass independently of body size and has been used to explore the connections between hydration status, symptoms, and survival in persons with advanced cancer [16,17]. By converting BIVA measurements to z-scores, researchers can compare body composition across different study populations, considering variables like cancer type, stage, gender, and ethnicity [18,19].

PhA serves as an indicator of cell membrane integrity and water distribution inside and outside the cell membrane [20]. It typically ranges from 5 to 7 in healthy populations but tends to decrease with age due to muscle mass loss and declining body fluid proportions [21]. PhA has been linked to increased mortality and morbidity in various patient groups and has demonstrated prognostic value in malignancies and chronic diseases, including CRC and PC [22,23,24].

However, in persons with advanced cancer, factors like dehydration or ascites can lead to imprecise BIA-derived FFM measurements. Discrepancies have been observed in advanced LC and CRC patients [3]. Additionally, BIA may inaccurately gauge FFM or FM in patients with breast and gynecological malignancies due to lymphedema [1].

To ensure the most accurate assessment, it is essential to determine whether BIA is being used as an indicator of nutritional and metabolic health or to assess risk based on derived body composition measurements [10]. Several studies have underscored BIA’s role in predicting preoperative risk and postoperative complications, identifying patients with sarcopenia in routine clinical practice, and providing valuable information on body composition changes in patients with cancer [5,16,25,26,27,28].

Therefore, BIA has demonstrated itself as a precise and valid tool with substantial clinical relevance, offering valuable insights that could have a significant impact on clinical outcomes [29,30,31].

This scoping review aims to evaluate the current scientific and clinical evidence concerning the use of BIA for assessing body composition in cancer patients undergoing active treatment.

## 2. Materials and Methods

### 2.1. Search Strategy

A literature search was conducted through MEDLINE (via PubMed), CINAHL (via EBSCO), Scopus (via Elsevier) and Web of Science databases, for all published articles until 3 July 2021.

Full search strategies are represented in Appendix A. This scoping review was conducted in accordance with the following research question “What is the clinical relevance of BIA as a valid tool to assess body composition in persons with malignancies for the adult population?” and followed the PCC mnemonic (The Joanna Briggs Institute Reviewer’s Manual), considering population: patients aged 18 years or older with cancer; concept: use of BIA to assess body composition; context: antineoplastic treatment (chemotherapy, radiotherapy or other).

Search followed PRISMA extension for scoping reviews (PRISMA-ScR). Search terms included Neoplasm OR Malignant Neoplasm OR Tumor OR Cancer OR Malignancy OR Malignancies AND Electric Impedance OR Electrical Impedance OR Bioelectrical Impedance OR Bioelectrical impedance analysis OR BIA OR Bioelectric Impedance OR Electrical Resistance AND Body Composition. All studies extracted from databases were imported into a systematic review tool—Rayyan QCRI 0.1.0 (Qatar Computing Research Institute, Doha, Qatar) software, and duplicates were removed. Subsequently, two reviewers independently screened the titles and abstracts of included studies using Rayyan and both inclusion and exclusion criteria. Conflicts were discussed and resolved between the reviewers. After full-text screening, study selection was conducted by one reviewer. All study designs written in English, regarding all cancer types and published within the last 10 years from 2011 to 2021 were considered.

### 2.2. Inclusion and Exclusion Criteria

The following inclusion criteria were used: (1) studies reporting body composition evaluation using Bioelectrical Impedance Analysis (BIA), (2) studies considering patients with cancer and (3) studies including participants ≥ 18 years. Studies including non-cancer pathology or only assessing nutritional status were excluded. After selection process according to titles and abstracts, full texts were evaluated.

### 2.3. Data Extraction

One reviewer extracted relevant data from the included studies and systematically organized them according to the following parameters: cancer location, author(s)/year of publication, country, study design, study objective, sample size, gender, BMI, mean age, BIA device, BIA frequencies, BIA equation, study limitations and main conclusions (Appendix B).

## 3. Results

A flow diagram of the included studies is represented in Figure 1. Out of 2070 total search results, 975 were duplicates. After screening all titles and abstracts, 1037 studies were excluded. A total of 58 articles were sought for retrieval, 7 of which were not retrieved because the full articles were unavailable (*n* = 1), or only available as conference abstracts (*n* = 4) or written in a foreign language (*n* = 2), resulting in 51 articles assessed for eligibility. Following full-text review, 15 articles were excluded (Appendix C) due to only focusing on PhA (*n* = 8), only assessing nutritional status (*n* = 3), including not only oncological patients, but also surgical patients (*n* = 2), focusing on fluid administration (*n* = 1) or including multiple diseases (*n* = 1). This scoping review included a total of 36 studies: 18 prospective studies [5,9,16,25,26,27,28,30,31,32,33,34,35,36,37,38,39,40], one retrospective study [41], six cross-sectional studies [29,42,43,44,45,46] and 11 systematic reviews [1,2,7,10,15,47,48,49,50,51,52].

### 3.1. Study Characteristics

The studies included in this review were published from 2012 to 2021 and had been conducted in different countries: China (3) [25,32,43], USA (3) [5,7,42], Netherlands (1) [9], Finland (3) [2,16,47], Portugal (2) [2,47], Sweden (2) [16,36], Canada (3) [1,42,49], Republic of Korea (4) [30,31,33,41], Poland (5) [27,37,38,46], United Kingdom (5) [10,26,34,50,51], Japan (1) [28], Australia (1) [34], Denmark (3) [35,44,48], Norway (2) [40,48], Brazil (1) [36], Germany (3) [29,39,45], Greece (1) [15], Italy (1) [39] and United Arab Emirates (1) [15].

### 3.2. Population

The population size for the included original articles (*n* = 25) ranged from 18 [35] to 1217 participants [39], comprising a total of 3015 patients with cancer (58.6% male and 41.4% female), with a mean baseline BMI that ranged from 20.3 to 30.0 kg/m^2^ and mean age that ranged between 47 and 70 years. Moreover, three studies included only female participants [33,42,46], and five studies included patients with HNC, i.e., nasopharyngeal cancer [5,32], oropharyngeal cancer, sinonasal cancer, salivary gland cancer and glottic cancer [5], squamous cell carcinoma in the oral cavity, oropharynx, hypopharynx or larynx [9,16,25]. Furthermore, three studies included patients with BC [33,42,46], one with EC [49], two with hepatocellular carcinoma [27,30], one with PC (ampulla of Vater carcinoma, pancreatic ductal carcinoma [28]), one with GC (adenocarcinoma, signet ring cell carcinoma [43]), seven with CRC [31,34,35,36,37,40,41], one with LC (non-small cell lung carcinoma [44]), one with SK (malignant melanoma [29]), one which included PC, GC and CRC [38] and two which included numerous cancer types (gastrointestinal, gynecological, head and neck, lung and pleura, genito-urinary, hematological, neuroendocrine, adrenal, thyroid, skin soft tissues, central nervous system and others) [39,45].

Review articles included in this scoping review (*n* = 11) comprised a total of 273 studies, ranging from 12 [10,47] to 42 articles [50], with a total sample of 78,350 participants. Of these, three studies considered patients with HNC [2,15,47], one with BC [48], one with EC [49], one with PC [50], one with GC [51] and four which included multiple cancer types (head and neck, lung, breast, gastric, esophageal, hepatocellular, pancreatic, colorectal, gynecological (including ovarian and endometrial), prostate and hematological malignancies) [1,7,10,52].

### 3.3. Body Composition Assessment: BIA

Appendix D and Appendix E resume the information collected from original research articles (*n* = 25) and review articles (*n* = 11) included in this review. From the 25 original articles comprised, 13 had exclusively used BIA as a method to assess body composition [16,25,26,27,28,30,32,33,37,38,39,41,46]. Eight studies had chosen both BIA and CT [5,29,31,34,36,43,44,45]; three selected BIA and DXA [9,40,42] and only one included BIA, MRI and skinfold thickness (ST) [35].

Moreover, 11 studies did not report the frequencies applied [5,25,27,28,29,32,35,39,41,43,46]; 10 used a frequency of 50 kHz [16,33,34,36,37,38,40,42,44,45]; one used frequencies of 50 and 1000 kHz [31]; one used frequencies of 5, 50 and 200 kHz [9]; one used frequencies of 0.5, 50 and 100 kHz [9] and one used frequencies of 1, 5, 50, 260, 500 and 1000 kHz [30]. Only five studies referred BIA equations used: two mentioned Geneva equation [9,40]; one mentioned Janssen equation [36]; two mentioned Sun et al. equation [39]. and one had used the equation in the BIA software [28].

Appendix F comprises information on BIA methodologies regarding BIA measurements, position of participants during assessment, position of participants during assessment, specific body composition measures assessed and raw BIA measures.

## 4. Discussion

This scoping review examines the current evidence regarding the use of BIA for body composition assessment in cancer and its implications for patient outcomes and prognosis of long-term survival.

### 4.1. Head and Neck Cancer

Persons diagnosed with HNC often face a heightened risk of malnutrition due to the location of the tumor and the impact of oncological treatments on their ability to consume food. This challenge is particularly pronounced among those with oral, oropharyngeal, and hypopharyngeal cancers. Consequently, there is a need for tools to identify malnutrition in this specific patient group.

Almada-Correia et al. [2] conducted a comprehensive review of the existing literature on body composition assessment in patients with HNC to determine the most appropriate method for this population. Their review considered various methods for assessing body composition in clinical settings, including anthropometry, BIA, CT, and DXA. The findings indicated that CT and DXA were the established standards for evaluating body composition in patients with cancer, although they are not routinely used in the management of HNC cases [2].

Another review [47] investigated changes in body composition among patients with HNC to identify the most effective methods for assessing body composition. This study underscored the significance of body composition evaluation and concluded that both skinfold thickness (ST), BIA, DXA, and CT demonstrated substantial reductions in lean body mass (LBM), FFM, and skeletal muscle mass (SMM), along with increases in body FM, among HNC patients undergoing chemoradiotherapy [47]. Importantly, it highlighted the regular availability of BIA, which holds promise for assessing body composition in patients with HNC.

Mantzorou et al. [15] summarized and discussed clinical data on the effectiveness of assessment tools, such as BIA, in evaluating malnutrition in persons diagnosed with HNC. The authors recommended additional studies to explore the role of BIA-derived measures in assessing the nutritional status of these patients [15]. Notably, there are only a limited number of studies that have examined the use of BIA in HNC patients.

Malecka-Massalska et al. [54] also emphasized the relevance of BIA, particularly BIVA, as a method that could provide objective measures to enhance clinical decision-making and predict outcomes in HNC patients. Similarly, Axelsson et al. [55] highlighted three different factors derived from BIA variables: Fat-Free Mass Index (FFMI), PhA and Standardized Phase Angle (SPA), adjusted for age and sex. A PhA cutoff value at 5.95° was identified as the most accurate predictor of 5-year survival. Both PhA and SPA were considered valuable prognostic tools for patients with advanced HNC [55]. In a prospective study conducted by Ding et al. [32], the study aimed to assess the changes in body composition among patients with nasopharyngeal carcinoma undergoing chemoradiotherapy. The findings of this study highlighted the significance of the BIA-derived FFMI in diagnosing malnutrition. The decrease in FFMI was found to be associated with a decline in the patients’ quality of life (QoL). As a result, the authors emphasized the importance of incorporating BIA into nutritional assessments [32].

In another prospective study [5], which sought to determine whether BIA could effectively identify sarcopenia in HNC patients, the primary outcomes revealed a robust correlation between BIA measurements and CT-based estimates. BIA demonstrated the ability to accurately identify sarcopenia, particularly among male patients, with a high level of sensitivity and specificity (>90%). This reinforced the practical utility of BIA in clinical settings for identifying patients with sarcopenia [5].

Additionally, Jager-Wittenaar et al. [9] conducted a study to evaluate the validity of BIA in patients with HNC and concluded that it serves as an acceptable tool for assessing FFM in clinical practice.

Two other studies [16,25] focused on describing BIA and assessing the correlation of BIA parameters with complication rates and other relevant indicators in patients with HNC. The first study [16] emphasized the utility of BIA parameters such as PhA and BIVA as valuable screening tools that provide essential information about body composition. The second study [25] highlighted BIA’s role as a clinically valuable tool in preoperative risk assessment, contributing to the reduction in complications and hospital stays.

Collectively, it is evident that BIA represents a cost-effective, non-invasive, rapid, and user-friendly approach for estimating the nutritional status of patients with HNC. It can play a crucial role in identifying patients who require nutritional care before the initiation of treatment, potentially reducing complications and hospital stays.

### 4.2. Breast Cancer

BC has emerged as the predominant cause of cancer incidence globally. The development of BC has been associated with the accumulation of adipose tissue in adulthood over the years. Recommendations for patients with BC include weight loss for those who are obese and a reduction in adiposity reserves.

In these patients, undetected or unaddressed malnutrition can lead to severe adverse outcomes. The existing body of literature demonstrates that malnutrition is linked to elevated morbidity and mortality, and the nutritional status plays a pivotal role in the prognosis of BC, potentially influencing the progression of the disease [56].

Nutrition-related symptoms, such as nausea, vomiting, loss of appetite, constipation, diarrhea, stomach pain, altered taste perception, sore mouth, and difficulty swallowing, can arise from the tumor itself or as side effects of treatment. These symptoms adversely affect dietary intake and elevate the risk of malnutrition. Malnourished patients with BC can tolerate fewer treatment cycles, rendering them more vulnerable to treatment-related toxicities and increased hospitalizations [56,57].

The impact of body composition on BC risk and outcomes in BC survivors is well-documented. Therefore, the integration of body composition assessment into the comprehensive care of these patients is essential. A scoping review involving persons with BC [48] aimed to investigate changes in weight and body composition, suggesting the need for further investigations with long-term prospective designs and consistent assessment of weight and body composition using the same measurement tools. BIA was mentioned as a cost-effective and user-friendly tool that could contribute to standardizing measurements [48].

In a cross-sectional study, Bell et al. [42] compared the accuracy of previously published Single Frequency-BIA equations predicting FFM with DXA measurements of FFM in a group of patients diagnosed with BC undergoing treatment. The study found that BIA consistently overestimated FFM. However, the authors acknowledged the need for future studies to develop and validate BIA prediction equations specifically tailored to BC patients [42].

Terada et al. [58] focused, as previously mentioned [1], on the limitations of single-use BIA in detecting lymphedema and measuring patient-reported outcomes. Similarly, in a study by Blaney et al. [59], BIA was capable of identifying 80% of true lymphedema cases but produced false negatives for 20%. Consequently, the authors recognized that BIA might not be a suitable tool for assessing established lymphedema [59].

Jung et al. [33] explored changes in weight, body composition, and physical activity in BC patients undergoing adjuvant chemotherapy, concluding that BIA could provide more concrete and objective results. Wilczyński et al. [46] reported that BIA does not cause ionization and is considered a gold standard in the field of body composition analysis.

### 4.3. Oesophageal Cancer

Somewhat similarly to HNC, patients with EC are at nutritional risk due to the location of the disease and side effects of anticancer therapies and/or surgery. Consequently, most patients with EC become malnourished. Thus, these patients represent a group of persons with cancer who are nutritionally compromised, due to dysphagia and oncological treatments. In addition, patients undergoing surgery for cancer are at particular risk of post-operative complications [60]. BIA may offer an additional method of identifying patients at risk of post-operative morbidity.

A recent review [49] on current literature regarding the assessment of body composition in EC patients could not agree on the best tool, due to inconsistencies in methods of assessing and reporting body composition, although authors recognized its usefulness regarding decision-making support in patients with EC.

Powell et al. [26] conducted a study with the objective of establishing a connection between low muscle volume (LMV) defined by Bioelectrical Impedance Analysis (BIA) in patients undergoing EC surgery and clinical outcomes. The findings of this study revealed that BIA-derived LMV served as a crucial prognostic indicator in patients undergoing potentially curative esophagectomy for cancer [26].

In a study by Ida et al. [61], the aim was to determine the impact of neoadjuvant chemotherapy (NAC) on the body composition of patients with EC and to assess the relationship between changes in body composition and the occurrence of postoperative complications. BIA-derived measurements such as BCM, FFM, and SMM exhibited significant reductions in patients who experienced postoperative complications. The study underscored the value of nutritional assessment through body composition in predicting postoperative complications following NAC in EC patients [61].

Similarly, after employing BIA to evaluate body composition, Motoori et al. [62] reported similar findings, concluding that the loss of SMM during NAC represented a significant risk factor for postoperative infectious complications in patients with EC undergoing esophagectomy.

Conversely, a different outcome emerged in the study by Miyata et al. [63], which aimed to assess changes in body composition using BIA during NAC for EC. According to their study [63], pre-NAC sarcopenia was not linked to the occurrence of postoperative complications, and a significant reduction in SMM was not observed in all persons diagnosed with EC [26,61,62,63].

### 4.4. Hepatocellular Cancer

Although common, malnutrition is frequently an underdiagnosed condition in patients with hepatocellular carcinoma (HCC). These patients are at a special increased risk for malnutrition as the liver is the central organ involved in nutrients metabolism [64].

In a prospective study conducted by Lee et al. [30], the impact of various factors, including Bioelectrical Impedance Analysis (BIA)-derived PhA, the presence of sarcopenia, and edema index, was evaluated in relation to postoperative complications in patients with hepatocellular carcinoma (HCC). The study found that BIA offered valuable additional clinical insights into the occurrence of postoperative complications in patients with HCC scheduled for surgery. However, it should be noted that due to a relatively short follow-up duration, the precise role of BIA in predicting short-term survival remained unclear [30].

In another study by Pagano et al. [65], the potential of BIA as a valuable tool for chronic liver diseases was highlighted. The study concluded that BIA-derived PhA had the capacity to predict the severity of liver disease and reflect the nutritional risk in HCC patients.

A cross-sectional study by Peres et al. [66] also recognized the relevance of BIA-derived PhA as an essential tool for nutritional assessment in the context of chronic hepatitis, liver cirrhosis, and HCC [30,65,66].

### 4.5. Pancreatic Cancer

PC has a poor overall prognosis, with a low 5-year survival rate. However, patients with early tumor resection have a higher chance of more successful treatments. Patients with PC often are malnourished and suffer from cancer cachexia. PC has been characterized as a highly catabolism inducer, with rapid depletion of the host’s body compartments. In fact, a large proportion of these patients have already lost 10% of body weight at the diagnosis. Early detection of wasting is central in the clinical approach of these patients [67]. Therefore, it is crucial to early assess nutritional status to identify and treat malnutrition and also to prevent or counteract cachexia. In clinical practice, anthropometric methods have been used but are not ideal: they are time-consuming and difficult to perform, especially in bed-ridden patients. Additionally, objective indicators such as serum albumin and transferrin are difficult to interpreter due to non-nutritional factors [50].

Regarding specifically PC, the only available systematic review on this topic was conducted by Bundred et al. [50], who summarized the existing literature on body composition assessment in patients with PC and assessed its impact on perioperative outcomes and long-term survival. Despite the lack of consensus regarding optimal methodology, this review highlighted the need for standardized assessment of body composition, reinforcing its importance to support future decision-making in PC patients [50].

Mikamori et al. [28] conducted a study with the goal of utilizing BIA for postoperative nutritional assessment in patients who had undergone pancreaticoduodenectomy. Their findings indicated that BIA proved to be a valuable tool for assessing body composition and nutritional status in patients who had undergone surgery for PC and other related conditions.

Gupta et al. [24] investigated the correlation between BIA-derived PhA and overall survival in patients with stage IV PC. Their study suggested that BIA-derived PhA served as a more robust predictor of survival compared to nutritional indices such as albumin, pre-albumin, and transferrin in patients with advanced PC.

In a similar way, Yasui-Yamada et al. [68] reported parallel findings concerning the significance of PhA. They highlighted PhA as a valuable prognostic marker for both short-term and long-term postoperative outcomes in patients with hepatobiliary-pancreatic cancer.

### 4.6. Gastric Cancer

GC leads to important depletion of muscle and fat tissue due to surgical interventions, chemo- and radiotherapy. In line with this, GC patients are at high nutritional risk and malnutrition-related to malignant disease, leading to lower compliance with treatment and complications during surgery. Complete surgical resection remains the only curative modality for early-stage GC. These patients in the perioperative period first consume LBM, which might not be evident from BMI nor other nutritional scores. BIA can overcome these difficulties [69].

In a cross-sectional study conducted by Gao et al. [43], the accuracy of BIA in estimating visceral fat area (VFA) in individuals with GC was explored. The study’s findings revealed a significant correlation and satisfactory reliability between VFA measurements obtained by CT and BIA [43].

In a retrospective study by Yuet al. [70], it was demonstrated that preoperative low BIA-derived PhA was a predictor of the risk of overall and severe complications in patients with GC. This suggests the potential use of BIA in assessing the risk of postoperative complications, especially in elderly patients with GC.

A review conducted by Kamarajah et al. [51] regarding the current literature on body composition assessment in patients with GC highlighted the absence of a consensus on the optimal methodology. It also emphasized the need for establishing guidelines for body composition assessment in patients with GC. Dzierżek et al. [38] examined body composition and its changes in patients who had undergone surgeries for GC, PC, and CRC. The authors concluded that BIA was a straightforward and effective method for assessing body composition and its alterations in patients undergoing major surgery [38,43,51,70].

### 4.7. Colorectal Cancer

Excess body weight and metabolic alterations have been identified as risk factors for cancer of the colon-rectum. Patients with advanced disease stage can develop a wasting-like phenotype of nutritional status, and gastrointestinal cancers globally show high prevalence of malnutrition.

In a prospective study led by Jones et al. [34], the primary objective was to assess the agreement between BIA and MAMC in comparison to CT scans for the measurement of muscle mass and the identification of sarcopenia in patients with CRC. The study’s conclusion revealed that both BIA and MAMC were found to be inadequate methods for detecting reduced muscle mass in patients with CRC when compared to the measurements of CT-derived muscle mass at L3 [34]. Another study [31] explored the relationships between a single cross-sectional area of skeletal muscle at the lumbar region (L3) measured by CT and total body SMM assessed by BIA in primary CRC patients. Kim et al. [31] identified BIA as an alternative method to CT scans, offering a non-invasive and cost-effective tool for assessing body composition status, including SMM, in patients with CRC.

Palle et al. [35] examined the correlation between a single cross-sectional thigh MRI, SMM as a reference, and multi-frequency BIA-derived FFM in CRC patients who had undergone chemotherapy. This study considered BIA and thigh MRI as suitable alternatives to the more complex and expensive MRI due to their consistent and correctable errors [35]. Ræder et al. [40] aimed to validate two different BIA devices against DXA in persons with CRC. Their study concluded that FFM estimated from both BIA devices exhibited good agreement with DXA, as long as the appropriate equation was used [40]. However, Bärebring et al. [71] reached contrasting conclusions in their study, which aimed to validate the ability of BIA, compared to DXA, to assess changes in FFM in non-metastatic CRC patients. BIA yielded imprecise data on changes in FFM, regardless of the equation used, and was therefore not deemed a valid option for quantifying changes in FFM in patients with CRC.

In a retrospective study, Song et al. [41] delved into the relationship between body composition and the platelet-to-lymphocyte ratio (PLR) in patients diagnosed with CRC. They discovered that body composition indices, including fat and muscle indices measured by BIA, were associated with PLR in these patients and could be related to a poorer prognosis in CRC cases [41]. A study by Souza et al. [36] aimed to evaluate the agreement between CT and surrogate methods commonly applied in clinical practice. The study revealed that while physical examination demonstrated the best agreement for assessing low muscle mass, Patient-Generated Subjective Global Assessment (PG-SGA), BIA, and CT exhibited similar prognostic values for survival [36]. In another study, conducted by Szefel et al. [37], the purpose was to determine the usefulness of BIA in detecting and monitoring cancer cachexia (CC) in patients with CRC. The study successfully associated BIA with the identification of differences in body composition depending on the cancer stage and the advancement of CC [37].

Gupta et al. conducted two separate studies investigating the prognostic role of PhA in advanced CRC [23] and the association between BIA-derived PhA and Subjective Global Assessment (SGA) [72]. The findings from both studies recognized BIA-derived PhA as a prognostic and potential nutritional indicator in advanced CRC patients [23,31,34,35,36,37,40,41,71,72].

### 4.8. Lung Cancer

Hansen et al. [44] conducted a study to explore the agreement between BIA assessment of body composition and software analysis of CT scans in patients with non-small cell LC. The study’s conclusion indicated that both methods were not directly comparable for body composition measurements. Furthermore, BIA was found to overestimate SMM and underestimate FM [44]. In another study, Kovarik et al. [73] provided a summary of recent evidence concerning various methods, including BIA, for assessing changes in body composition in LC patients. The study underscored the importance of BIA and BIA-derived PhA in predicting outcomes for LC patients. Similarly, Gupta et al. [74] observed that BIA-derived PhA served as an independent prognostic indicator in patients with stage IIIB and IV non-small cell LC [44,73,74].

### 4.9. Skin Cancer

A prospective study conducted by Zopfs et al. [29] evaluated the correlation between simple, planimetric measurements in CT slices and measurements of patient body composition and anthropometric data, performed with BIA and metric clinical assessments. This study concluded that simple measurements in a single axial CT slice could determine body composition parameters, with high clinical relevance [29].

### 4.10. All Cancer Types

Adequate body composition has been proved essential for neoplastic disease outcome. BIA has been found to be a prognostic indicator in several chronic conditions, including cancer [24].

Cereda et al. [39] conducted a study with the aim of investigating the potential independent prognostic roles of FFMI, BMI, and weight loss (WL), and their associations with quality of life (QoL) in a substantial cohort of patients with cancer. While acknowledging the versatility and non-invasive nature of BIA for bedside assessments, the study emphasized the necessity for additional confirmatory studies to validate the usefulness and prognostic value of BIA-derived FFMI in people diagnosed with cancer [39].

In a cross-sectional study by Mueller et al. [45], the validity of BIA as a diagnostic tool in patients with malignancies, both with and without malnutrition, was investigated. BIA was recognized as a valid diagnostic tool for assessing muscle and FM, and its use was recommended for the early detection and short-term follow-up of malnutrition and cachexia [45].

A review conducted by Di Sebastiano et al. [1] aimed to identify key considerations for body composition analysis in diverse populations with cancer and to discuss various methods of body composition assessment, such as anthropometric measures, BIA, ADP, DXA, CT, and MRI. The review deemed DXA as the optimal method for whole-body composition analysis in prospective use with cancer populations due to its precision, accuracy, and fewer limitations compared to other methods. The authors highlighted the limited applicability of BIA in advanced and BC patients, as it relies on water volume and cannot distinguish tumor or lymphedema within lean and fat tissue depots [1]. Aleixo et al. [75] summarized the existing literature on BIA’s role in assessing sarcopenia in adults with cancer. BIA was considered an accurate method for detecting sarcopenia and a viable alternative to CT, DXA, and MRI in the field of oncology.

Grundmann et al. [7] conducted a review of the current scientific and clinical evidence regarding the utility of BIA in patients with malignant neoplasms. The study noted that BIA and PhA provided valuable information for assessing nutritional and overall health status in patients diagnosed with cancer. While acknowledging the need for further research, the authors encouraged the use of BIA in clinical practice [7]. A recent study by Di Vincenzo et al. [76] found that BIA-derived PhA was decreased in patients with sarcopenia, suggesting PhA as a valuable parameter for detecting low muscle quality and identifying sarcopenia. Małecka-Massalska et al. [52] described the existing literature on the utility of BIA in assessing cancer-related malnutrition. The study considered BIA to be an objective, reliable, and non-invasive method for assessing malnutrition [52].

A systematic review conducted by Matthews et al. [10] aimed to assess whether BIA measures and estimates of body composition could identify adults with oncological diseases at risk of complications. The use of BIA in the peri-operative period was found to be useful in predicting the risk of complications following elective cancer surgery [10].

### 4.11. General Considerations for the Use of BIA

Appendix G focused on BIA’s advantages and disadvantages, as a method for estimating body composition in patients with cancer.

Bioelectric impedance analysis (BIA) is a safe [49], non-invasive [1,2,7,15,47,50,51,52], easy-to-use [1,2,15,50,51], reproducible [1,2], indirect method [2] for measuring body composition. It appears to show good correlations, when compared to other methods [47], despite being considered less accurate than radiological assessment methods [49].

Although it has good application consistency and is considered a useful tool for assessing the nutritional status of persons with cancer [2,52], BIA can underestimate FFM in patients with advanced cancer, compared with DXA [1]. Moreover, considering BIA depends on water volume, it has limited use in people with advanced-staged cancer and patients with BC and is not capable of distinguishing tumor or lymphedema in the lean and fat tissue depots [1]. While BIA is recognized as a validated tool to assess body composition in patients with cancer [2,47,52], some studies reported some inconsistent findings, with poorer accuracy and precision in obese and edematous individuals [50,51]. As a practical and objective assessment method [1,50,51,52], BIA is also considered to be relatively inexpensive, when compared to more sophisticated methods like DXA, CT or MRI [1,2,7,47,49], although it is usually more expensive than anthropometric measures [1]. Often reliable [52], one of BIA’s disadvantages resides in the fact that it is unable to measure the entire body, which results in incomplete information [2].

BIA is a portable [1,7,15,47], time and cost-effective technique [15], that requires little training [1]. However, this tool is not routinely available outside the research setting [49]. Many prediction equations using linear regression rely on BIA to estimate body composition, based on variables that may differ between different populations and were derived from healthy individuals [1,2,15,47]. Nonetheless, BIA-derived PhA, a prognostic factor of patient survival [15], which, along with BIVA, is considered to reflect both nutritional and hydration status [7,15], does not depend on regression equations to be calculated [47]. BIA-derived measures (FFM, FM, body weight, BMI) are correlated with an increased risk of developing colon cancer and potentially other cancers [7] and may serve as early indicators for improvement in nutritional and health status [7,52], but can also be useful to evaluate and predict outcomes, such as post-operative complications [10].

Despite the many advantages associated with BIA use in general and specifically in patients with cancer, the literature presents a multiplicity of equipment used to measure BIA, methods and participant preparation protocols, which in turn can lead to different readings obtained by BIA. Therefore, evaluations should consider an adequate fluid balance and food intake and be performed under the same circumstances, taking into consideration possible sources of error. Moreover, its efficiency may be affected by nutrition status, physical activity, phase of the menstrual cycle, placement of electrodes, limb length, blood chemistry, altered fluid balance, oedema, endocrine diseases, treatment with growth hormone, acute illness, intensive care treatment, organ transplantation, position of the body and movements during the measure, type of electrodes, use of oral contraceptives [1,2]. This method can also be compromised by loss of accuracy when patients are in the extremes of BMI ranges (≤16 kg/m^2^ or ≥35 kg/m^2^) [2,47].

### 4.12. Limitations

The overall strength of conclusions from this review is limited by the heterogeneity of the studies, different objectives, BIA devices and methodologies, and findings regarding BIA’s assessment of body composition in patients with oncological diseases. Furthermore, the initial literature search was only conducted in four different databases and research articles were only considered when written in English, which may have excluded other potentially relevant studies. Different cancer locations were taken into consideration in this review, making it challenging to condense information. Also, the limited number of existing studies by cancer location, included in this review, highlights the need for further studies.

Additionally, this scoping review did not perform a quality evaluation on included articles, which may also be a limitation of this review. Overall findings drawn from this scoping review must be interpreted carefully, considering different types of study designs included.

## 5. Conclusions

The identified studies have considered BIA to be a suitable and valid method for the assessment of body composition in oncology. BIA-derived measures have shown good potential and relevant clinical value in preoperative risk evaluation, in the reduction of postoperative complications and hospital stay and as an important prognostic indicator in patients with cancer. Hence, research encourages the implementation of this method in the nutritional assessment at a larger extent.

Future research focusing on the diagnostic value and clinical applications of BIA and BIA-derived PhA should be conducted in order to increase knowledge and strengthen evidence on its impact on patient survival and other clinical outcomes, including postoperative complications and treatment-related toxicity, that justify the increase of its use in clinical practice.

## Figures and Tables

**Figure 1 nutrients-15-04792-f001:**
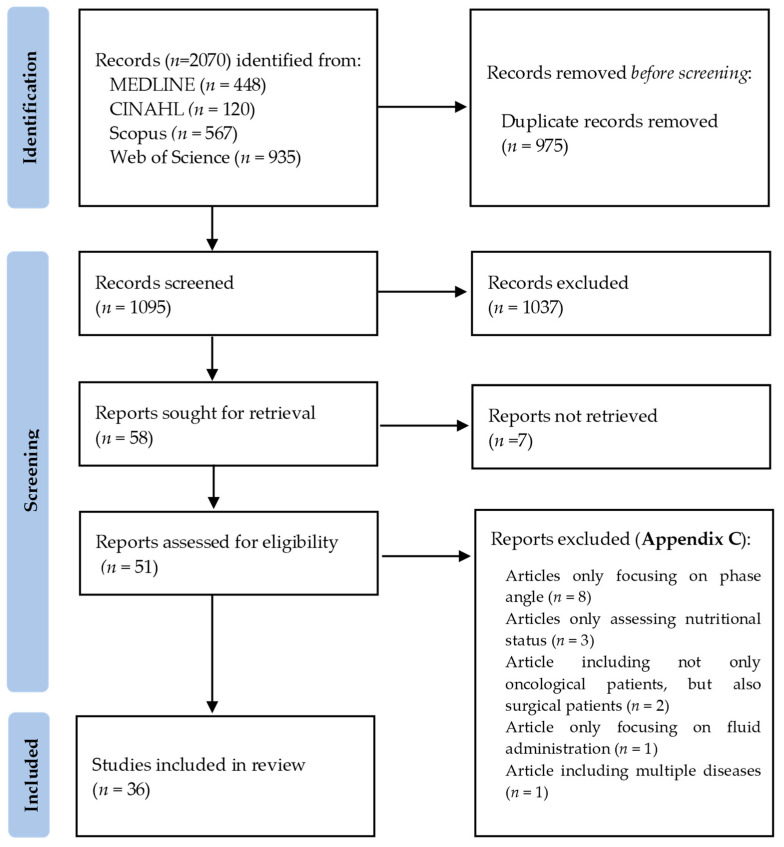
Flow diagram of included studies, according to PRISMA 2020 Guidelines [53].

## Data Availability

Not applicable.

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
