# Peer review of "Bioelectrical Impedance Analysis (BIA) for the Assessment of Body Composition in Oncology: A Scoping Review"

_nutrients, 2023, doi:10.3390/nu15224792_

Round 1

Reviewer 1 Report

Comments and Suggestions for Authors

The manuscript under the title Bioelectrical Impedance Analysis (BIA) for the Assessment of Body Composition in Oncology: a scoping review is an interesting and thorough manuscript in the use of BIA in Oncology. 

The only concern I have is that by including reviews and systematic reviews in the analysis it is a possibility of repeating the results of the studies already mentioned. Therefore, I would suggest excluding the reviews from your analysis and use them only to detect studies that were not retrieved in your search. 

Author Response

Reviewer 1:

We thank you for the time and effort that you dedicated on our manuscript, and we are grateful for the insightful comments to improve our paper.

Please see below, in blue, for a point-by-point response to the comments and concerns.

  1. The manuscript under the title Bioelectrical Impedance Analysis (BIA) for the Assessment of Body Composition in Oncology: a scoping review is an interesting and thorough manuscript in the use of BIA in Oncology.

Authors response: We thank so much the Reviewer for the positive feedback.

  1. The only concern I have is that by including reviews and systematic reviews in the analysis it is a possibility of repeating the results of the studies already mentioned. Therefore, I would suggest excluding the reviews from your analysis and use them only to detect studies that were not retrieved in your search.

Authors response:  We thank you the suggestion. However, a scoping review is not to show the evidence but the state of the art. All the scoping reviews (namely the JBI which is used by us) methodologies allow the inclusion of SR. Therefore, the criteria for scoping reviews are not the same for another SR.

As this is a scoping review, we have included all available studies focusing on the topic. This is to ensure that the existing views in the literature (including conclusions presented in systematic reviews) are included and discussed in our paper. Of note, the existing scoping reviews do include systematic reviews in their methodology. Joanna Briggs Institute Reviewers’ Manual: 2015 edition/supplement states this: “For the purposes of a scoping review, the “source” of information can include any existing literature, e.g. primary research studies, systematic reviews, meta-analyses, letters, guidelines, etc.”

Reviewer 2 Report

Comments and Suggestions for Authors

Thank you for the opportunity to review the article titled “Bioelectrical Impedance Analysis (BIA) for the Assessment of Body Composition in Oncology: a scoping review”. This scoping review maps the existing research reporting on the use of BIA for body composition assessment of patients with active cancer. While the subject matter addressed is indeed pertinent and timely, the article exhibits certain deficiencies that could potentially undermine its scientific rigor and overall research impact.

Foremost among these concerns is the absence of comprehensive data extraction from the incorporated literature concerning the use and validity of the BIA methodology. For instance, the authors have provided detailed descriptions of the BIA devices in terms of their model/brand, manufacturer, used frequencies, and equations. However, the results section neglects to include other information of critical importance to readers. This includes, for example, whether the BIA devices are employed for the measurement of whole-body or segmental body composition, the position of participants during assessment (i.e., supine or standing), participant preparation protocols (e.g., fasting requirements, exercise restrictions), the specific body composition measures assessed (e.g., fat mass, fat-free mass, skeletal mass), and the raw BIA measures. Furthermore, the authors have reported overarching conclusions regarding the validity of BIA from the included articles but have failed to support these statements with data, which has the potential to lead to erroneous interpretations.

In addition, the discussion section requires refinement to transform it into a comprehensive discussion on the primary findings derived from the scoping review, as opposed to a mere recapitulation of the individual articles included therein. Finally, the way the incorporation of systematic review articles contributes to the results of the scoping review remains somewhat unclear.

A minor, yet noteworthy, concern pertains to the presentation of references within the text. The absence of parentheses or subscript markers for cited references can pose challenges for readers attempting to distinguish between data and citation sources.

Comments on the Quality of English Language

No major issues identified.

Author Response

Reviewer 2:

We thank you for the time and effort that you dedicated on our manuscript, and we are grateful for the insightful comments to improve our paper.

Please see below, in blue, for a point-by-point response to the comments and concerns.

  • Thank you for the opportunity to review the article titled “Bioelectrical Impedance Analysis (BIA) for the Assessment of Body Composition in Oncology: a scoping review”. This scoping review maps the existing research reporting on the use of BIA for body composition assessment of patients with active cancer. While the subject matter addressed is indeed pertinent and timely, the article exhibits certain deficiencies that could potentially undermine its scientific rigor and overall research impact.

Authors response: We thank the Reviewer for the positive feedback.

  • Foremost among these concerns is the absence of comprehensive data extraction from the incorporated literature concerning the use and validity of the BIA methodology. For instance, the authors have provided detailed descriptions of the BIA devices in terms of their model/brand, manufacturer, used frequencies, and equations. However, the results section neglects to include other information of critical importance to readers. This includes, for example, whether the BIA devices are employed for the measurement of whole-body or segmental body composition, the position of participants during assessment (i.e., supine or standing), participant preparation protocols (e.g., fasting requirements, exercise restrictions), the specific body composition measures assessed (e.g., fat mass, fat-free mass, skeletal mass), and the raw BIA measures. Furthermore, the authors have reported overarching conclusions regarding the validity of BIA from the included articles but have failed to support these statements with data, which has the potential to lead to erroneous interpretations.

Authors response: Thank you for pointing this out. We have now added information regarding these issues in the revised version of our manuscript (please see line 207-209). All the information is comprised of a new table: Appendix VI.

  • In addition, the discussion section requires refinement to transform it into a comprehensive discussion on the primary findings derived from the scoping review, as opposed to a mere recapitulation of the individual articles included therein.

Authors response: We have now modified the Discussion section as requested.

  • Finally, the way the incorporation of systematic review articles contributes to the results of the scoping review remains somewhat unclear.

Authors response: We thank you the suggestion. However, as this is a scoping review, we have included all available studies focusing on the topic. This is to ensure that the existing views in the literature (including conclusions presented in systematic reviews) are included and discussed in our paper. Of note, the existing scoping reviews do include systematic reviews in their methodology. Joanna Briggs Institute Reviewers’ Manual: 2015 edition/supplement states this: “For the purposes of a scoping review, the “source” of information can include any existing literature, e.g. primary research studies, systematic reviews, meta-analyses, letters, guidelines, etc.”

  • A minor, yet noteworthy, concern pertains to the presentation of references within the text. The absence of parentheses or subscript markers for cited references can pose challenges for readers attempting to distinguish between data and citation sources.

Authors response: We thank the Reviewer for this valuable suggestion and have now changed the reference format to follow the instructions given by Nutrients (reference numbers are placed in square brackets [ ], and placed before the punctuation).

Round 2

Reviewer 2 Report

Comments and Suggestions for Authors

The authors have addressed most of my concerns, except the following “The authors have reported overarching conclusions regarding the validity of BIA from the included articles but have failed to support these statements with data, which has the potential to lead to erroneous interpretations.” I would expect to see some sort of data in Appendix IV, such as data on sensitivity, specificity, systematic effect, agreement). Instead the authors simply summarized the main conclusions of the articles, which could lead to misleading interpretations on the BIA validity by readers.

Minor comments:

Use person-first language (e.g., patients with cancer, patients with HNC) throughout the manuscript.

Appendix VI: Abbreviations have not been used consistently in the table. Please revise the table to include only abbreviations, and a footnote spelling them out.

Please double check if all abbreviations were used consistently throughout the manuscript.

Lines 63-79: The authors have not included any references to support their statements. Same for lines 147-157, 177-183, 199-201, 257-258. Also, what is the link between malnutrition and body composition assessment? The authors discussed the impact of malnutrition in different conditions within these new lines but have not linked malnutrition and body composition in the introduction.

Line 205-206: What does a ‘standard BIA equation’ mean? Ref 28 seems to be wrongly placed in line 205.

Author Response

Reviewer 2:

We thank you for the time and effort that you have dedicated on our manuscript, and we are grateful for your insightful comments to improve our paper.

Please see below, highlighted in blue, our point-by-point responses to your comments and raised concerns.

  • The authors have addressed most of my concerns, except the following “The authors have reported overarching conclusions regarding the validity of BIA from the included articles but have failed to support these statements with data, which has the potential to lead to erroneous interpretations.” I would expect to see some sort of data in Appendix IV, such as data on sensitivity, specificity, systematic effect, agreement). Instead the authors simply summarized the main conclusions of the articles, which could lead to misleading interpretations on the BIA validity by readers.

Authors response: We thank the Reviewer for the observation. We have included to Appendix IV data on BIA sensitivity and specificity.

Minor comments:

  • Use person-first language (e.g., patients with cancer, patients with HNC) throughout the manuscript.

Authors response: We thank the Reviewer for this suggestion and have now modified the revised version of our manuscript accordingly.

  • Appendix VI: Abbreviations have not been used consistently in the table. Please revise the table to include only abbreviations, and a footnote spelling them out.

Authors response: We thank the Reviewer for this suggestion and have now modified the revised version of the table accordingly.

  • Please double check if all abbreviations were used consistently throughout the manuscript.

Authors response: We thank the Reviewer for pointing this out. We have now revised the manuscript to include all abbreviations.

  • Lines 63-79: The authors have not included any references to support their statements. Same for lines 147-157, 177-183, 199-201, 257-258. Also, what is the link between malnutrition and body composition assessment? The authors discussed the impact of malnutrition in different conditions within these new lines but have not linked malnutrition and body composition in the introduction.

Authors response: We thank the Reviewer for this comment. We have now included references for those topics.

We also included information about the link between malnutrition and body composition in the Introduction, as follows:

Nutritional deterioration and progressive unintentional weight loss are prevalent conditions in patients diagnosed with cancer,and are widely associated with poor outcomes and complications through the course of the disease and treatments [1,2].

Cancer leads to metabolic alterations that contribute to depletion in nutritional status resulting in changes in body composition, which in turn are negative predictors of  therapy toxicity, clinical outcomes, quality of life and survival. Therefore, it is critical to timely identify and treat malnutrition in order to enhance clinical outcomes. Thus, body composition should be part of nutritional assessment in these patients. Nevertheless, it remains a challenge due to a variety of methods and tools to assess nutritional status and body composition in patients with cancer [1].

  • Line 205-206: What does a ‘standard BIA equation’ mean? Ref 28 seems to be wrongly placed in line 205.

Authors response: We thank the Reviewer for pointing this out. We have now corrected the sentence accordingly.
